# Exploring the Relationships between Safety Compliance, Safety Participation and Safety Outcomes: Considering the Moderating Role of Job Burnout

**DOI:** 10.3390/ijerph18084223

**Published:** 2021-04-16

**Authors:** Xiaoyi Yang, Boling Zhang, Lulu Wang, Lanxin Cao, Ruipeng Tong

**Affiliations:** School of Emergency Management and Safety Engineering, China University of Mining and Technology-Beijing, Beijing 100083, China; yangxyi6@126.com (X.Y.); zbl9856@163.com (B.Z.); wanglulu1027@126.com (L.W.); caolanxinlx@163.com (L.C.)

**Keywords:** safety compliance, safety participation, safety outcomes, job burnout, occupational psychological health

## Abstract

Safety compliance (SC) and safety participation (SP), which are key factors predicting safety outcomes (e.g., accidents, injuries and near misses), are related but distinct. However, which component is more significant remains controversial. Job burnout is a typical occupational psychological syndrome of employees that impacts safety outcomes, but the role that it plays in the relationship between SC, SP and safety outcomes is ambiguous. To clarify these relationships, Chinese coal mine workers were sampled. Then, hypotheses on the influencing mechanisms were initially proposed and later tested empirically. To conduct this testing, SC and SP scales were revised, and a job burnout scale was developed. The results showed that there were significant relationships between workers’ SC and SP and safety outcomes; meanwhile, exhaustion, cynicism and low professional efficacy had significant effects on these relationships. Job burnout acted as a significant and indispensable moderator. Moreover, workers’ occupational psychological health condition should be considered when improving safety outcomes.

## 1. Introduction

Safety performance traditionally consists of safety compliance (SC) and safety participation (SP) [1,2]. The former refers to the core activities that employees need to perform for the purpose of maintaining workplace safety, for example, wearing personal protective equipment correctly. The latter refers to voluntary behaviors of individuals that could contribute to developing a safety-supportive environment rather than directly ensuring personal safety, for instance, helping coworkers [1]. Through meta-analyses and cross-sectional designs, it has been reported that both these components can contribute to improving safety outcomes (e.g., accidents and injuries) [3,4,5]. However, what emerges from these existing works is the arguments regarding which component has a stronger relationship with safety outcomes. For example, Christian et al. [4] reported no significant difference between SC and SP in terms of their relationship with accidents and injuries, but Clarke [3] pointed out that SP had a stronger influence on safety outcomes than did SC. In contrast to Clarke [3], DeArmond et al. [5] argued that the interrelationship between SC and safety outcomes was more significant than SP.

Furthermore, some strain factors may impact the interaction between SC, SP and safety outcomes. The causes that undermine individuals’ energetic resources to reach work goals are strain factors [6,7]. Among strain factors, job burnout is a pure example and is the most representative psychological syndrome associated with work. It has been discussed widely in various fields, and most frontline workers suffer from it [8,9]. Thus, when investigating the mechanisms through which SC and SP affect safety outcomes, the role of burnout is of high interest and significance. However, research has barely explored the interaction between SC, SP, safety outcomes and job burnout.

To that end, frontline coal miners in China, who work in a typical high-risk and labor-intensive industry and whose behavioral safety management is a key issue [10,11,12], were selected as participants. First, the relationships among the abovementioned factors were identified. Then, a series of measuring instruments were revised or developed, and surveys conducted. Finally, an in-depth analysis was performed employing a structural equation model (SEM), and the implications of this research discussed. Additionally, some concrete and specific measures are provided to promote the management effectiveness of individuals’ behavior.

## 2. Literature Review and Hypothesis Development

### 2.1. SC, SP and Safety Outcomes

Based on job performance theory [13,14], Neal et al. [1] introduced the model of safety performance and identified the two components, namely, SC and SP. The validity of these two distinct forms of safety performance has been widely proven, and they have been increasingly adopted in studies [2,15]. Reviewing them, we can condense some points. On the one hand, SC and SP are generally proposed as mediators, based on the independent and dependent variables, and their interactions and mechanisms, which are investigated. On the other hand, these two components are usually treated as outcome variables, with the antecedents and their relationships being explored. Safety climate and safety leadership are the most important predictors, and the former significantly promote SC and SP [2,3]. The latter, which predominantly focuses on the styles of transaction and transformation, can also contribute to employees’ SC and SP [16,17]. Regarding the safety outcomes that these two components influence, attention is mainly paid to accidents, injuries and near misses, and SC and SP, as noted, can effectively reduce the frequencies and rates of these undesirable results. Furthermore, we focused on accidents, injuries and near misses regarding safety outcomes in this work.

After thoroughly reviewing previous studies, we propose two hypotheses, which are listed below. The grounds are as follows. First, as noted above, it has been shown that better SC and SP can predict a low accident rate, less injury and fewer near misses in work. Thus, Hypotheses 1 and 2 may be proposed.

**H1:** *SC is negatively associated with safety outcomes of accidents, injuries and near misses*.

**H2:** *SP is negatively associated with safety outcomes of accidents, injuries and near misses*.

### 2.2. The Role of Job Burnout

Job burnout is the prolonged reaction of individuals to chronic emotional and interpersonal stressors in their job, and it is traditionally defined as a psychological syndrome of employees consisting of three dimensions: exhaustion, cynicism and low professional efficacy [18,19]. To depict burnout, exhaustion describes its basic individual stress dimension, in which a person feels overextended and depleted of his or her emotional and physical resources. Cynicism describes the interpersonal dimension of burnout, in which a person responds to his or her job with a negative, hostile or excessively detached attitude, commonly accompanied by a loss of idealism. Finally, low professional efficacy describes the self-evaluation dimension of burnout, in which a person feels that his or her competence and productivity regarding managing the work are declining.

It has been reported that most frontline manual laborers suffer from burnout, including coal miners [20] and construction workers [21]. Research has shown that occupational burnout can create negative health outcomes of workers [22,23], such as hypertension [24], sleep problems [25], anxiety [26] and others. In particular, as special occupational groups, most frontline manual laborers are at a low education level and their social status is low; working in a special environment of high temperature, high pressure or dust for a long time caused them to languish and burnout. Lu et al. [27] found that nearly 85.98% of factory workers and miners experience occupational burnout. To ensure the physical and mental health of workers and improve work performance, it is important to reduce the job burnout of workers.

According to the model of safety performance proposed by Neal and Griffin [28], compliance and participation are two dimensions of safety performance. Both compliance and participation are behaviors instead of cognition. From our perspective, it is clear that job burnout cannot be a mediator between either SC or SP and safety outcomes; hence, we may conjecture that burnout moderates the relationship between these two components and safety outcomes. Besides, job burnout would have a negative impact on the physical and mental health of coal miners, which may easily lead to unsafe behaviors, accidents and other safety outcomes [29]. Taking into account the previous assumption that safety compliance and safety participation have a negative impact on safety outcomes, Hypotheses 3 and 4 may be proposed. Finally, to summarize, we depict these relationships in a hypothetical model, as shown in Figure 1.

**H3:** *The three dimensions of job burnout, namely, exhaustion (H3a), cynicism (H3b) and low professional efficacy (H3c), have negative moderating effects on the relationship between SC and safety outcomes*.

**H4:** *The three dimensions of job burnout, namely, exhaustion (H4a), cynicism (H4b) and low professional efficacy (H4c), have negative moderating effects on the relationship between SP and safety outcomes*.

## 3. Materials and Methods

### 3.1. Measures and Instruments

#### 3.1.1. SC and SP

Regarding workers’ SC and SP, Neal et al. developed a comprehensive scale to measure these two components [1,2]. Many related studies in various occupations applying the initial scales or the revised version have been conducted. To suit the characteristics and working practices of frontline coal miners, as well as Chinese culture, the scales used in this paper were reworded and rephrased. First, the contents and substance of the SC and SP scales were obtained from previous questionnaires [1,5,11,30] and other reports, and both SC and SP initially included six items. Second, we discussed the draft with 12 squad leaders employed as frontline safety supervisors in charge of safety in their production team and five full-time safety inspectors who supervise frontline miners (including the squad leaders) to ensure readability and face validity; all of these respondents worked at a state-owned coal mine located in Shanxi, China. Third, based on the information collected from the coal mine, all the items were weighted, and essential revisions were performed, such as deleting some items, replacing some words or giving some examples to clarify some expressions. All the items were measured on a 5-point Likert scale ranging from 1 (strongly disagree) to 5 (strongly agree). Fourth, a sample of 250 frontline miners from the same coal mine was selected to conduct a pilot survey, which aimed to examine the quality and availability of the scales. We received 216 responses (for a response rate of 86.4%), of which 203 were valid responses (for an effective response rate of 94.0%). Finally, some items were slightly altered again, and items whose item-to-total correlation was less than 0.50 were removed based on our examination of reliability and correlations, as shown in Table 1.

#### 3.1.2. Job Burnout

Regarding job burnout, the Maslach Burnout Inventory (MBI) is the most popular and widespread measuring instrument [31]. In general, there are three distinct versions of the MBI, and five specific versions can be differentiated [32]. Specifically, the initial versions were developed for healthcare professionals (Maslach Burnout Inventory–Human Services Survey, MBI–HSS) and teachers (Maslach Burnout Inventory–Educators Survey, MBI–ES), and then, the Maslach Burnout Inventory–General Survey (MBI–GS) was developed for general occupations. Later, the MBI–HSS and MBI–GS were further revised for medical personnel (MBI–HSS(MP)) and students (MBI–GS(S)). Additionally, other valuable scales have been developed, such as the two-dimensional OLBI (Oldenburg Burnout Inventory) [33] the one-dimensional scales of BM (Burnout Measure) [34] and the S–MBM (Shirom–Melamed Burnout Measure) [35].

The MBI–GS has been used in different occupations to measure job burnout, but it was confirmed that this scale, which was a perfect foundation, should be further developed for specific occupations, especially for high-risk industries [36]. Thus, focusing on coal miners in China, we developed the job burnout scale based on the MBI–GS.

First, the symptoms of these workers’ job burnout were discussed with 12 squad leaders and 10 frontline miners, and each discussion lasted for 1.5 to 2 h. Then, we also showed and discussed the three dimensions of burnout and the items of the MBI–GS. The respondents’ work experience ranged from 6 to 15 years, with an average of 9.5 years, and all of them had an educational level from junior high school to high school or technical secondary school. Thus, on the one hand, the work experience of the coal miners could guarantee their in-depth understanding of this particular underground operation. On the other hand, their educational level could represent the average educational attainment of this group, and they could grasp the intention of this discussion and provide some constructive suggestions.

Finally, regarding job burnout, we obtained 43 symptoms and 12 suggestions to improve the items of the MBI–GS. After further analysis and classification of these results, studies confirmed that exhaustion, cynicism and low professional efficacy cover these symptoms well. However, some revisions needed to be made to make the items more readable and intelligible for frontline miners due to the gaps in the different cultural backgrounds and linguistic expressions between China and the English-speaking countries in which the items were first developed. Therefore, we initially developed a job burnout scale for coal miners that contained 18 items. Two items from the BM, MBI–HSS and OLBI scales were extracted and revised to cover all the burnout symptoms of frontline miners in the Chinese context. All items were also measured on a 5-point Likert scale.

To further analyze the scale, similar to SC and SP, a sample of 250 frontline miners was selected to conduct a pilot survey, and we received 216 responses (for a response rate of 82.4%), of which 197 were valid (for an effective response rate of 91.2%). Then, exploratory factor analysis (EFA) was conducted with SPSS 25.0. Specifically, the factor-loading matrix was obtained through principal component analysis. To ensure the appropriateness of the principal component analysis, the value of KMO (Kaiser–Meyer–Olkin) was 0.892. The methods employed by Yang et al. [36] were adopted to revise the initial scale.

The results showed that 5, 6 and 5 items were retained for exhaustion, cynicism and low professional efficacy, respectively. Thus, we obtained a job burnout scale for coal miners with a total of 16 items that explained 83% of the total variance. Regarding the reliability index, we obtained Cronbach’s alpha (α), which was 0.871, and the rotated component matrix, presented in Table 2.

#### 3.1.3. Safety Outcomes

Following McCabe et al. [37], He et al. [38] and Hu et al. [39], safety outcomes of accidents, injuries and near misses were measured. Specifically, the participants were asked to recall their experiences related to these safety outcomes in the past three months. The questions related to their safety outcomes in work were asked, “How many work-related accidents/injuries/near misses have you been involved in your past three months?”. A list related to these safety outcomes in coal mine were provided to help participants give appropriate responses, and the association between injuries and accidents were avoided. The results obtained were standardized to a 5-point scale based on the total number of three safety outcomes, with 1, 2, 3, 4 and 5 points meaning never, once, two to three instances, four to five instances and more than five instances, respectively.

### 3.2. Participants

The survey was administered to frontline miners in a state-owned coal mine in Shanxi, China. A total of 450 questionnaires were distributed; 406 responses were received, for a response rate of 90.2%, of which 367 were valid responses, for an effective response rate of 90.4%. All of the participants were male, almost all of them were younger than 50, and their educational level was mainly junior high school and high school or technical secondary school. Regarding the characteristics related to their work, most of them had worked more than five years, and their duties mainly included dig-in and coal mining. All the details are shown in Table 3.

## 4. Results and Discussion

### 4.1. Descriptive Statistics

Confirmatory factor analysis (CFA) was employed to examine whether or not the data fit the hypothetical model. As shown in Table 4, the factor loadings and the Cronbach’s alpha (α) exceeded 0.7. Hence, we can judge that the factors and dimensions included in our hypothetical model are reliable. In addition, the composite reliability (CR), average variance extracted (AVE) and discriminant validity were also reported, which are shown in Table 5. The values of CR and AVE exceeded 0.7 and 0.5; thus, we can conclude that the convergent validity and discriminant validity were demonstrated.

The levels of SC, SP, job burnout and safety outcomes are also illustrated in Table 4. The coal miners’ safety incidents had an average of 1.20, which was consistent with the fact [39]. However, the level of near misses, which we defined as incidents that “may result in injury, illness, damage or death but did not”, was relatively high with an average of 3.22, implying that the number of instances of safety incidents in the last three months was approximately three times.

The coal miners’ job burnout was at a very high level. According to Maslach et al. [32], exhaustion and cynicism above 2.70 and 1.80 are high; in our work, these two values were 3.03 and 2.34, respectively. For professional efficacy, a value below 3.30 is low, and for the coal miners in this study, we observed that it was 1.89, which was rather low.

### 4.2. Hypothesis Testing and In-Depth Analysis

SPSS AMOS Version 25.0 (Manufacturer: IBM Corporation in New York, the United States of America) was employed to conduct regression analysis to build the hypothetical model. The accepted model is presented in Figure 2, which reports the significance levels of the hypotheses. Furthermore, the fitness of the model was tested, as recommended by Kline [40], and the fit indices are shown in Table 6, which indicates a relatively high degree of fit [41].

#### 4.2.1. The Relationships between SC, SP and Safety Outcomes

As noted above, H1 and H2 proposed that the coal miners’ SC and SP should be negatively related to safety outcomes; the results strongly supported the hypothesis (β = −0.627, *p* < 0.001; β = −0.369, *p* < 0.001, respectively). This is because the level of safety compliance and safety participation reflects the safety awareness and attitude of workers, which has a vitally important impact on safety outcomes [42]. SC and SP are two distinct types of behavior. The former concerns employees’ in-role behavior and relates to task performance, which can be viewed as part of a person’s work role and predicts his or her material and spiritual rewards [2,3]. The latter concerns workers’ extra-role behavior and relates to contextual performance, which is more voluntary and can predict a safety-supportive environment [2,15].

In detail, given the characteristics of frontline workers in the high-risk and labor-intensive industries in which we are interested, i.e., coal mine workers in China, they are almost all from rural areas, have a low level of education and unskilled and inexperienced. Thus, it is common that these employees carry out core activities (SC) to a larger degree than voluntary activities (SP) when completing their scheduled work. Meanwhile, it is usually impractical and impossible to change the factors that may impact SC and SP in pure high-risk industry [43], such as high stress and a poor environment. Under this condition, both SC and SP will be undermined, and in their mindset, employees will give priority to in-role behavior rather than extra-role behavior. In addition, unsafe practices persist because they are “naturally” reinforced. That is, the fruits of taking shortcuts will be immediate and positive, for example, achieving a task with less effort or performing illegal operations to reduce time and save energy [2]. Hence, it is implied that the effect of SC on safety outcomes is more sensitive than that of SP; and SC is generally necessary and directly related to tangible (e.g., financial) or intangible (e.g., praise) rewards, especially the former, which these workers take more seriously.

#### 4.2.2. The Indispensable Moderator of Job Burnout

The results of H3 and H4 suggested that job burnout has negative moderating effect on the relationship between SC and safety outcomes, and the relationship between SP and safety outcomes. The significant interactional coefficients of each dimension (exhaustion, cynicism, low professional efficacy) of job burnout with the relationships between SC (β = 0.638, *p* < 0.001; β = 0.503, *p* < 0.001; β = 0.376, *p* < 0.001, respectively), SP (β = 0.661, *p* < 0.001; β = 0.511, *p* < 0.001; β = 0.387, *p* < 0.01, respectively) and safety outcomes demonstrate the moderating effects, which are presented in Figure 2. The miners whom we selected presented a very high level of burnout, and job burnout significantly undermined the positive effects of SC or SP on preventing their safety outcomes.

The reasons are as follows. First, as mentioned above, there are the environmental and organizational characteristics of this high-risk industry. As a pure psychological syndrome caused by various influencing factors, burnout is suffered by employees in almost every industry, and high-risk industries, such as the construction industry [21] and coal industry [20], are no exception. Second, job burnout itself is an “occupational phenomenon” and specifically involves phenomena in the occupational context [44]. Specifically, the long-term work and workplace factors in coal mines will also cause occupational phenomena in miners. That is, they will make them feel depleted of energy or exhausted, make them feel negativity or cynicism related to their job, increase their mental distance from the job and reduce their professional efficacy. This condition of miners will impact their work performance, especially their safety performance in a high-risk industry in which safety-related matters are essential. Regarding miners’ safety outcomes, which is also an unsafe practice can be predicted by SC and SP; its relationship with SC and SP is affected by this occupational phenomenon, namely, job burnout, which plays the role of a moderator in this relationship. Furthermore, because this occupational phenomenon is common and miners greatly suffer from it, it undermines the interactions of SC, SP and safety outcomes to the same degree, which also indicates the terrible situation of this phenomenon in miners.

### 4.3. Findings, Implications and Limitations

#### 4.3.1. Findings

This study explored the mechanisms through which SC and SP affect safety outcomes and the role of job burnout in these relationships. It was confirmed that coal miners’ job burnout symptoms were also perfectly covered by the three-dimensional structure of job burnout, namely exhaustion, cynicism and low professional efficacy. Additionally, four hypotheses in our study were all generally validated. Thus, we found that SC and SP will contribute to improving safety outcomes; however, job burnout will undermine this process. Moreover, the level of burnout that coal miners suffer is very high, which indicates that attention should be paid more to the occupational psychological health problems in this industry.

Further analyzing the results, first, we drew our samples from a high-risk industry, that is, the coal industry. Comparing with the construction industry, which is also a typical high-risk industry [45], there are several similarities between these two industries. On the one hand, regarding the characteristics of the work, in both industries, the work is complicated, dynamic, dangerous and labor intensive [46]. On the other hand, work related to safety is crucial in both of these two industries; hence, many efforts are made to ensure safety, such as building a positive safety climate and enhancing workers’ SC and SP. Furthermore, frontline employees in China, whether coal miners or construction workers, have similar characteristics; both groups of workers are mainly rural migrant workers with a low level of education, and their work is characterized by overloading, physical exhaustion and time [47]. Additionally, they inevitably suffer from various stressors, such as work pressure [20] and work–home conflict [48], which are the antecedents of their occupational psychological health problems in the workplace, for instance, job burnout.

Therefore, as mentioned above, coal miners suffer a high level of burnout, and construction workers experience the same problem. Other psychological health problems negatively affect employees in these two high-risk industries, such as psychological distress and pain [49] and depression [50]. Some studies have argued that these problems should be handled, particularly in Industry 4.0 [51], because these problems concern employees’ health and safety in the workplace [20,52]. Safety outcomes are also influenced by these problems. Additionally, there is an increasing emphasis on promoting safety outcomes based on the consideration of occupational psychological factors in other fields [53].

Based on this analysis, we offer some recommendations. First, not in the coal industry but in the construction industry, the occupational psychological health problems associated with frontline workers should be considered and investigated for the purpose of improving employees’ health and safety. Second, specific to the improvement of safety outcomes, SC and SP should undoubtedly be strengthened; additionally, other traditional measures that have been frequently discussed by studies and about which no details are necessary should be taken. More importantly, we wish to impel that the occupational psychological factors related to employees should be considered when improving safety outcomes; however, to the best of our knowledge, no enterprises in these two industries in China pay attention to this problem. Third, when occupational psychological factors are involved, management can follow the “environment/organization–occupational psychology behavior” process, which is more systematic and comprehensive [20].

#### 4.3.2. Implications

Theoretically, focusing on one high-risk industry (i.e., coal industry), we investigated the mechanisms through which SC and SP affect safety outcomes. Moreover, a pure psychological syndrome, which is also a widely discussed occupational psychological factor, was included to explore the role that it plays. Furthermore, the necessity of paying attention to employees’ occupational psychological health problems when implementing research related to safety management was proposed. In addition, the job burnout scale for Chinese coal miners was developed on the basis of the MBI–GS, which could serve as a reference for other studies.

Practically, in order to better ensure the progress of safe production work, managers should focus on the relationship between safety compliance and safety participation on safety outcomes. Besides, to conduct safety management in high-risk industries, based on the findings we obtained, it is not enough to simply consider general factors that may predict safety outcomes; it is important to pay attention to employees’ occupational psychological health condition. Moreover, the influence of job burnout on the relationship between safety performance and safety outcomes should be comprehensively considered in the management process. Therefore, we can cope with job burnout from three aspects to improve the occupational psychological health problems of workers. First, reasonable arrangement of work shifts is needed for workers to ensure that they have spare time to enjoy life. Second, it is necessary to provide occupational personal protection and health education and psychological counseling. Last, companies are obliged to improve the working environment.

#### 4.3.3. Limitations

There are some limitations in this study that need to be mentioned. First, regarding the samples we selected, only a coal mine in Shanxi provided the participants; hence, more samples in other provinces should be selected in future work to obtain more convincing and reasonable results. Second, only one high-risk and labor-intensive industry was sampled; hence, it is necessary to conduct research in other industries, such as construction. Therefore, the current summarized results need to be cautiously interpreted. Future research could be extended to include more areas of sampling and verified by research in other industries. Third, we performed our work using cross-sectional surveys; apart from this, a longitudinal study is necessary in future work.

Last but not least, the study was based on self-report measures, which increases the possibility of bias in general methods. Therefore, it is suggested that future research use multiple data sources to resolve this problem. For example, data could be obtained from the participant worker and the people who know the worker well (e.g., colleagues). Besides, the crux of subjective measurement lies in the difference in personal evaluation criteria. As a result, individual deviations and measurement errors may distort the final evaluation results. In order to solve this problem, future research should pay more attention to considering the use of objective measurements rather than subjective measurements alone.

## 5. Conclusions

This research delved into the relationships between SC, SP and safety outcomes; the role played by job burnout was also identified. It was discovered that safety outcomes cannot be effectively improved by focusing only on SC and SP. Furthermore, the mechanisms through which SC and SP affect safety outcomes were revealed. This study showed that SC and SP have a significant negative effect on safety outcomes. Meanwhile, job burnout acted as a moderator in these relationships; that is, it undermined the decreasing effect of SC and SP on safety outcomes.

Moreover, the very high level of job burnout influenced the interaction of SC and SP with safety outcomes at the same level, to some degree reflecting the terrible situation of frontline employees’ occupational psychological health problems in high-risk industries.

More in-depth, the terrible state of job burnout indicated that frontline workers’ occupational psychological health problems should be considered, and their occupational psychological factors should be involved when conducting safety management in high-risk industries. To that end, the environment/organization– occupational psychology-behavior process could be followed.

## Figures and Tables

**Figure 1 ijerph-18-04223-f001:**
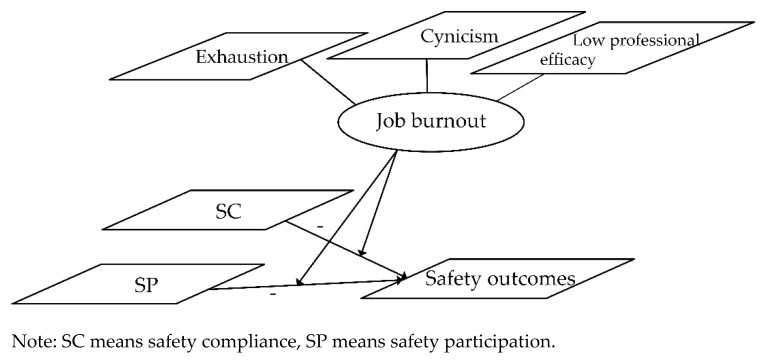
Hypothetical model.

**Figure 2 ijerph-18-04223-f002:**
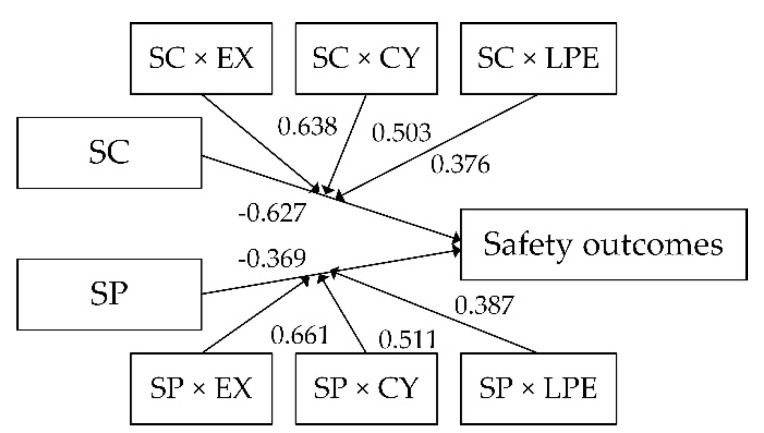
The accepted SEM.

**Table 1 ijerph-18-04223-t001:** The items of the SC and SP scales.

Components	Items	
SC	1–1.	I use all the required safety equipment during my working time, such as keeping on my gloves even if I feel that doing so is inconvenient
	1–2.	I comply with the necessary safety rules and procedures during my working time, such as the safety operating instructions for my post
	1–3.	I ensure the highest levels of safety during my working time, such as checking the environment to ensure safety
	1–4.	I take the appropriate steps if I was prevented from or punished for exercising my rights under safety rules and procedure, such as arguing with my squad leader
SP	2–1.	I help my coworkers make sure that they perform their work safely, such as taking action to stop violations
	2–2.	If I notice any safety-related matters, I always let my squad leader or safety inspector know the issues
	2–3.	I make extra efforts to make the safety of my work better, for example reforming the method the job is done to make it safer
	2–4.	I volunteered to carry out tasks or activities which could improve workplace safety, for example, attending safety-oriented but non-mandatory trainings frequently
	2–5.	I encourage my coworkers to work safely, such as communicating the results if an accident happens

Note: SC means safety compliance, SP means safety participation.

**Table 2 ijerph-18-04223-t002:** Rotated component matrix of the job burnout scale for coal miners.

Items		Dimensions		
		EX	CY	LPE
1.	My job makes me feel emotionally drained	**0.824 ^3^**	0.062	0.053
2.	I feel tired and fatigued after work	**0.893**	0.087	0.106
3.	When I weak up in the morning, once I notice I have to face my job, the feeling of exhausted exhaustion emerges	**0.871**	0.165	0.027
4.	Working all day is really a strain for me	**0.837**	0.302	0.089
5.	I am so weak and susceptible to illness ^1^	**0.763**	0.288	−0.097
6.	I always express negative emotions at work	0.221	**0.856**	0.032
7.	I have become more cynical about whether my work contributes anything	0.176	**0.695**	0.066
8.	I feel less and less interested in my job since I started as an employee	0.327	**0.747**	0.103
9.	The meaning of my job is doubtful	0.154	**0.833**	0.058
10.	My work bores me a lot	0.364	**0.847**	0.083
11.	I just want to finish the assigned job and not be disturbed by other coworkers or things ^2^	0.277	**0.712**	0.053
12.	I feel that I am making effective contributions to what my company does	−0.029	0.073	**0.708**
13.	I will feel comfortable when I complete a task effectively	−0.034	–0.127	**0.673**
14.	I am able to effectively solve the problems in my work	−0.031	0.266	**0.761**
15.	In my opinion, I am good at my job	0.003	0.173	**0.694**
16.	I feel exhilarated after I solve a problem in my work	0.234	–0.289	**0.685**

Note: EX, CY and LPE mean exhaustion, cynicism and low professional efficacy, respectively. ^1^ This item was exacted from the BM and OLBI scales. ^2^ This item was exacted from the MBI–HSS scale. ^3^ The bold means this Item match with the Dimension.

**Table 3 ijerph-18-04223-t003:** Demographic distribution of the participants (n = 367).

Characteristic	%	Characteristic	%
Gender		Work experience	
Male	100	Less than 5 years	21.2
Age		5–10 years	37.7
≤30	25.4	11–20 years	26.3
31–40	45.7	Most than 20 years	14.8
41–50	24.6	Trade type	
≥51	4.3	Dig-in	19.7
Education level		Coal mining	21.4
Primary school or below	10.3	Transportation	18.6
Junior high school	43.8	Electromechanical guarantees	16.5
High school or technical secondary school	38.5	Ventilation	8.2
Junior college or above	7.4	Other	15.6

**Table 4 ijerph-18-04223-t004:** Statistical results of the measure.

Items	Loading	α	M	SD	Items	Loading	α	M	SD
Safety performance					CY-1	0.736	0.785	2.227	0.601
SC-1	0.649	0.806	3.982	0.641	CY-2	0.664		2.074	0.572
SC-2	0.745		3.653	0.625	CY-3	0.827		2.523	0.532
SC-3	0.896		3.566	0.632	CY-4	0.674		2.302	0.544
SC-4	0.677		4.010	0.638	CY-5	0.776		2.431	0.556
SP-1	0.763	0.811	1.877	0.852	CY-6	0.683		2.510	0.562
SP-2	0.687		3.145	0.847	LEP-1	0.658	0.791	2.003	0.607
SP-3	0.661		2.664	0.823	LEP-1	0.761		1.762	0.668
SP-4	0.739		2.542	0.836	LEP-1	0.745		1.807	0.701
SP-5	0.724		2.566	0.859	LEP-1	0.893		1.941	0.683
Job burnout					LEP-5	0.762		1.934	0.674
EX-1	0.823	0.801	2.963	0.732	Safety outcomes				
EX-2	0.645		3.417	0.726	Accidents	0.756	0.821	0.132	0.010
EX-3	0.667		3.182	0.674	Injuries	0.861		0.266	0.033
EX-4	0.805		2.766	0.698	Near misses	0.963		3.225	0.074
EX-5	0.834		2.857	0.708					

Note: M means mean value, SD means standard deviation.

**Table 5 ijerph-18-04223-t005:** Results of convergent validity and discriminant validity examination.

	CR	AVE	Discriminant Validity					
			SC	SP	EX	CY	LEP	Safety Outcomes
SC	0.833	0.559	**0.748 ^1^**					
SP	0.840	0.512	0.568	**0.716**				
EX	0.871	0.576	0.421	0.352	**0.759**			
CY	0.871	0.532	0.543	0.326	0.574	**0.729**		
LEP	0.877	0.589	0.419	0.564	0.463	0.612	**0.767**	
Safety outcomes	0.898	0.747	0.602	0.477	0.546	0.493	0.467	**0.864**

Note: CR means composite reliability, AVE means average variance extracted. ^1^ The bold means discriminant validity.

**Table 6 ijerph-18-04223-t006:** Fit indices of the accepted structural equation model (SEM).

Index	GFI	RMR	RMSEA	AGFI	NFI	CFI	IFI
Result	0.886	0.059	0.068	0.884	0.889	0.882	0.887
Evaluation	Good	Moderate	Reasonable	Good	Good	Good	Good

Note: GFI means goodness-of-fit index, RMR means root mean square residual, RMSEA means root mean square error of approximation, AGFI means adjusted goodness-of-fit index, NFI means normed fit index, CFI means comparative fit index, IFI means incremental fit index.

## Data Availability

Not applicable.

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
