# Peer review of "Exploring the Relationships between Safety Compliance, Safety Participation and Safety Outcomes: Considering the Moderating Role of Job Burnout"

_ijerph, 2021, doi:10.3390/ijerph18084223_

Round 1

Reviewer 1 Report

Dear authors,

First of all, thanks for submitting your work to be considered in IJERPH. After a careful read of your manuscript, I would like to highlight the relevance of the topic, and the interesting topic addressed in this study.

The paper has, however, certain issues that I believe should be amended by you, in order to improve it and increase its scientific value and relevance. Please see below:

  • The literature review is good in terms of describing the basic concepts used in the study, but poor at empirically supporting how does burnout relate to workers’ (physical and mental) health and performance issues, especially in the case of vulnerable workforces, needing to strengthen the description of these mechanisms in order to support the study hypotheses.
  • Also, and as authors mention job stress as a key issue at the section “results”, it is a bit awkward not having a good (although brief) conceptualization on its close relationship to burnout at the literature review.
  • Only as a suggestion, authors could refer to (among others) empirical sources such as: 10.1017/S1138741600003863 and 10.1371/journal.pone.0211447 to cover the aforementioned issues.
  • Many statistical operators and symbols were misused along the text. For instance, the intext description of some test results are clumsily described, including unusual issues, such as capitalized “P” values and mean values not preceded by the statistical operator (M).
  • Table 5 is good, but the qualitative interpretation of the cut-off points requires more background (and support) at the methods section.
  • From, my opinion, the biggest non-methodological shortcoming of this work is the fact that it lacks from a well-developed discussion. Although authors were considerably organized and systematic to describe the results, the latter should be firstly (and briefly) presented, and then the study hypotheses could be contrasted in the glance of the existing literature. In its present form, readers may struggle to identify what was a study-related finding and what corresponds rather to an inference of the authors on the basis of external sources. My suggestion is to re-organize these two sections, being clearer and more systematic, and to develop the discussion directly in the glance of the study hypotheses.
  • The limitations of the study were poorly addressed. A third party reading throughout the manuscript could easily remark a further list of critical limitations that were not properly acknowledged by the authors. Moreover, the limitations included were not properly discussed, describing (e.g.) how could them impact the validity and reliability of the results, what were the remedies considered by the authors, and how to improve similar further studies.
  • The insights allowing to link this study with both occupational and public health remain quite implicit. In other words, one could assume that the study itself is valuable, but it results very difficult to identify the concrete value of the conclusions beyond some associations and further needs identified by the authors in pages 11 and 12 of the manuscript. Please consider elaborating and discussing more on the practical value and concrete hints provided buy this study.

I hope my comments may result useful for the authors during their revisions of the paper.

Best wishes.

Author Response

On behalf of my co-authors, we are truly grateful to yours and other reviewers’ positive and constructive comments and suggestions on our manuscript entitled “Exploring the Relationships Between Safety Compliance, Safety Participation and Safety Outcomes: Considering the Moderating Role of Job Burnout”. We have studied reviewers’ suggestions carefully and have tried our best to revise our manuscript according to the comments. We have uploaded our point-by-point responses to the reviewers’ comments in doc version as the “Author Responses to Reviewer Comments-ijerph-1142207-R1” file. For your convenience, this doc document can reflect our responses and revisions more clearly.

Reviewer 2 Report

Coal miners are a high-risk industry. How to reduce the dangerous accidents of coal miners is very important for this industry. This phenomenon also shows the importance and value of this research. However, based on academic requirements, I made the following review comments.

  1. The biggest crux of this research lies in the role of “job burnout” in the research structure. The authors believe that job burnout belongs to the role of moderator. In other words, job burnout will cause the impact of SC and SP. However, SC and SP are cognitive-level problems, and job burnout is a manifestation of explicit behavior. Logically, it should be the cognitive level that affects the performance of explicit behavior. In other words, in theory, SC and SP should affect job burnout. Therefore, Job burnout should be the role of mediator. If the authors still believe that job burnout belongs to the role of moderator, please add the inference process and related literature (Supplement line 102-109).

  1. In line 77-85, the authors hope to illustrate the importance of SC and SP through the comparison of regression coefficients, and propose H3 accordingly. This approach is inappropriate. The verification of data required for quantitative research is not just a comparison of numbers. Therefore, unless the authors use rigorous statistical verification methods to test the importance of SC and SP, the assumption of H3 is unnecessary. It is suggested that the author cancel the hypothesis of H3, and use discussion to illustrate the importance of SC and SP after H1 and H2 are established. This phenomenon also appears in the hypothesis of H6 (line 107-109). The authors are requested to amend or provide explanations.

  1. In line 110-115, please add the positive or negative assumption of moderating effect (if the authors still insist that job burnout is a moderator), and explain and discuss in line 107-109.

  1. The presentation of Figure 1 and Figure 2 ask the authors to consider whether to adjust. As far as the structural equation model is concerned, an ellipse (or circle) is generally used to represent the latent variable, and the square represents the observed variable.

  1. In line 138-139, authors use a 5-point scale in measurement process. According to Krosnick, Judd and Wittenbrink (2005), the 7-point scale of the questionnaire will help improve the quality of data. Therefore, recent studies have emphasized the measurement specifications of the 7-point scale. Please explain why the 5-point scale is still used in the scale design.

  1. In line 226-229, the CFA of SEM checks the loading of the item, not the loading of latent variables. Table 4 presents the loading of latent variables. The authors are requested to delete this part and provide the factor loading value of each item (this is the basic requirement of SEM).

  1. In line 227-228, the authors use a Cronbach’s alpha threshold of 0.6. This standard is too loose. It is recommended that authors use a Cronbach’s alpha threshold of 0.7.

  1. In line 230-232, the author uses the average of SC and SP to illustrate "SC is the core activity for ensuring safety during work". This conclusion may not be rigorous. Since the factors that affect the average also include sample distribution and other issues, there is not much information about the average. It is recommended that authors not make too much explanation on the average.

  1. The title of Table 4 is "This is a table. Tables should be placed in the main text near to the first time they are cited." This is an obvious error. The authors should correct it.

  1. In Table 4, the authors combined the three sub-dimensions of job burnout into one Cronbach’s alpha to show that it is inappropriate and unnecessary. Because the authors consider the three sub-dimensions of job burnout to be examined separately in subsequent research, it is important to tell readers about the Cronbach’s alpha of the three sub-dimensions. In addition, the value of "Mean" and "SD" should show three digits after the decimal point.

  1. The entire article should be processed in the SEM method. May I ask why the authors did not propose the values of CR, AVE, and discriminant validity? These values are the basic requirements of SEM.

  1. How do the authors deal with the statistical problem of moderation? Because Amos is more complicated in operating moderation, and different scholars have put forward different views. If the authors use Amos as a moderation analysis tool, please indicate which scholar’s analysis method is used.

  1. In Figure 2, the authors mark the standardized regression coefficients on the graph and add a significant asterisk after the standardized regression coefficients. This way of expression is wrong because the significance of the regression coefficient is not calculated from the standardized regression coefficient. Please correct it.

  1. The fit indices in Table 5 are all within the acceptable range. It is recommended that authors delete Table 5 and describe fit indices in line 244-248.

  1. In line 268, "2" should be an extra number, please delete it.

  1. In line 287-288, "Second, unsafe practices are "naturally" reinforced, and the fruits are immediate and positive." This sentence is not easy to understand, and the authors should explain it again.

  1. In Table 3, “N=367” should be expressed as “n=367”.

  1. "p" should be lowercase and italic.

Author Response

(The authors gave the same response as above.)

Reviewer 3 Report

The authors are commended for this interesting study on an important occupational safety and health issue among a unique cohort of workers.  The study aims are clearly stated, the methods are appropriate to the aims, and the results well articulated.  

The only improvement I can suggest is to discuss briefly how these results will be disseminated and integrated into workplace safety and health programs.

A final review of English usage might be helpful but the paper is clear in its present form.

Author Response

(The authors gave the same response as above.)

Round 2

Reviewer 1 Report

Dear Authors,

First of all, thanks for addressing my previous comments and suggestion with all the possible rigor and seriousness.

Through a careful read of this revised manuscript, I have checked that all my concerns were properly responded and resolved by the authors, putting an acceptable level of detail in their responses and several amendments in the revised version of the paper, to my satisfaction*.

*Only in case the paper needs a further round of revisions from authors, I would suggest them to improve the links presented between burnout and psychological health, since this issue was just scarcely addressed in their R1 version of the paper, and more evidences are required to strengthen and support this important and hazardous relationship for occupational health practitioners.

Therefore, my opinion is that the paper can be accepted for publication in IJERPH.

Best wishes and good work!

Reviewer 2 Report

The authors have made substantial revisions to the previous manuscript, but there are still two problems that need to be corrected. I make the following comments.

1. In line 2, 14, 17-18, 40, 47, 250, 281-282, 393, “between” should be revised to “among”. This is a grammatical issue.

2. In line 113-120, the H3 and H4 are problematic (including the representation in Figure 1). Because job burnout is divided into three sub-dimensions, which are part of the authors' investigation, when discussing the literature, the authors did not know that job burnout would be divided into several sub-dimensions before performing EFA analysis. In other words, in the literature discussion, the authors can only assume the moderator role of job burnout, but cannot make assumptions about the sub-dimensions of job burnout. After completing the EFA analysis, you can perform analysis based on the EFA results. This is a logical problem. The authors are requested to make corrections.